# LOPT: Learning Optimal Pigovian Tax in Sequential Social Dilemmas

**Yun Hua**[1]*, **Shang Gao**[2]*, **Wenhao Li**[3], **Haosheng Chen**[2], **Bo Jin**[3],
**Xiangfeng Wang**[4,5,6]†, **Jun Luo**[1], **Hongyuan Zha**[7]

[1] Antai College of Economics and Management, Shanghai Jiao Tong University
[2] School of Computer Science and Technology, East China Normal University
[3] School of Computer Science and Technology, Tongji University
[4] Key Laboratory of Mathematics and Engineering Applications (MoE)
[5] Shanghai Institute of AI for Education, East China Normal University
[6] Shenzhen Loop Area Institute (SLAI)
[7] School of Data Science, Chinese University of Hong Kong (Shenzhen)
{hyyh28,jluo_ms}@sjtu.edu.cn, {shanggao,hschen}@stu.ecnu.edu.cn
{bjin,whli}@tongji.edu.cn, xfwang@cs.ecnu.edu.cn, zhahy@cuhk.edu.cn

## Abstract

Multi-agent reinforcement learning (MARL) has emerged as a powerful framework for modeling autonomous agents that independently optimize their individual objectives. However, in mixed-motive MARL environments, rational self-interested behaviors often lead to collectively suboptimal outcomes situations commonly referred to as social dilemmas. A key challenge in addressing social dilemmas lies in accurately quantifying and representing them in a numerical form that captures how self-interested agent behaviors impact social welfare. To address this challenge, *externalities* in the economic concept is adopted and extended to denote the unaccounted-for impact of one agent's actions on others, as a means to rigorously quantify social dilemmas. Based on this measurement, a novel method, **L**earning **O**ptimal **P**igovian **T**ax (**LOPT**) is proposed. Inspired by Pigovian taxes, which are designed to internalize externalities by imposing cost on negative societal impacts, LOPT employs an auxiliary tax agent that learns an optimal Pigovian tax policy to reshape individual rewards aligned with social welfare, thereby promoting agent coordination and mitigating social dilemmas. We support LOPT with theoretical analysis and validate it on standard MARL benchmarks, including Escape Room and Cleanup. Results show that by effectively internalizing externalities that quantify social dilemmas, LOPT aligns individual objectives with collective goals, significantly improving social welfare over state-of-the-art baselines.

## 1 Introduction

Reinforcement learning [42] achieved remarkable efficacy across diverse domains [32, 21, 18, 52] and has been successfully extended to multi-agent settings, especially in fully-cooperative scenarios [46, 26, 49]. Nevertheless, prevalent centralized multi-agent reinforcement learning (MARL) methods that utilize team rewards [13, 41, 38, 37, 7] are encumbered by inherent limitations in their scalability to large agent populations and are fundamentally deemed unsuitable for self-interested agents in mixed-motivation environments. While decentralized learning paradigms [43, 40, 2], wherein agents independently optimize their individual rewards, provide a more natural modeling approach for self-interested behavior. Yet, these methods frequently encounter difficulties in facilitating coordination

---

*Equal Contribution.
†Corresponding to: Xiangfeng Wang.

among agents. In many real-world environments with mixed motives—particularly those involving exclusionary or subtractive common-pool resources [36, 22, 23]—rational, self-interested behavior often leads to collectively suboptimal outcomes. These situations are known as *social dilemmas*.

The concept of social dilemma, originating from economics, refers to situations in which individually rational decision-making leads to collectively suboptimal outcomes [19]. Specifically, these scenarios arise when mutual cooperation would generate universal benefits, yet agents are incentivized to defect due to the prospect of greater individual gain from non-cooperative behavior. In the context of mixed-motivation MARL, social dilemmas are formally characterized as conflicts between individual reward maximization and the optimization of joint or collective returns [22]. This framing reflects a core economic insight: strategies that are rational from an individual agent's perspective can produce inefficient or undesirable outcomes at the group level. Accordingly, a central challenge in mixed-motivation MARL research is the development of theoretically grounded mechanisms to accurately quantify and represent social dilemmas in a numerical form that captures how self-interested agent behaviors impact social welfare. This involves not only assessing the long-term influence of self-interested agent policies on social welfare but also designing learning algorithms capable of aligning individual incentives with collective welfare over extended time horizons.

Established economic theory has long applied the concept of externalities to explain social dilemmas [44, 8, 6]. An externality arises when the actions of one economic agent directly affect the utility or production possibilities of others without these effects being accounted for in market transactions [30]. These impacts on individual utility or production capacities collectively contribute to changes in social welfare. Positive externalities arise from actions that benefit social welfare, while negative externalities result from actions that harm it. Based on this theoretical foundation, many policy instruments—both market-based and non-market-based—have been developed to mitigate the negative impacts of externalities and resolve corresponding social dilemmas [35, 1, 5]. A prominent example is the Pigovian tax/allowance [5, 28], such as carbon tax *, which levies taxes on any market activity that generates negative externalities and provides allowances to which bring positive externalities [34], thereby incorporating these effects into market prices. This process known as externality internalization.

Inspired by economic theory, we introduce externality into mixed-motivation MARL to address the challenge of accurately quantifying and representing social dilemmas in a numerical form. This theoretical perspective offers a clear way to describe MARL dilemmas, where an agent's actions may impact others without those effects being reflected in its own reward—creating externalities. Building on this insight, we further propose a learning-based solution that leverages Pigovian tax/allowance mechanisms to alleviate these issues by subsidizing behaviors with positive externalities and taxing those with negative ones. The proposed method, **L**earning **O**ptimal **P**igovian **T**ax (**LOPT**), introduces a centralized agent, referred to as the **tax planner** which learns to allocate tax and allowance rates by maximizing the long-term global reward. This learning process is proven to be equivalent to approximating the optimal Pigovian tax, which reflects the value of externalities. The learned rates are then used to design a novel reward shaping mechanism, termed optimal Pigovian tax reward shaping, which shapes each agent's local reward to reflect how its actions impact overall social welfare. Compared to existing handcrafted or performance-driven reward shaping methods for addressing social dilemmas, LOPT offers a theoretically sound shaping approach based on optimal Pigovian tax, which is computed by optimizing social welfare. This ensures theoretical guarantees while demonstrates superior empirical effectiveness and adaptability.

The primary contributions of this paper are as follows:

- **Externality theory is introduced in MARL** to quantify and represent social dilemmas numerically, providing a theoretically grounded framework for capturing the impact of self-interested agent behaviors on social welfare.

- **A centralized tax/allowance mechanism based on reward shaping LOPT is proposed** to approximate the optimal Pigovian tax and internalize the externalities of self-interested agents in mixed-motivation MARL tasks, thereby aligning individual agent incentives with social welfare and addressing social dilemmas.

---

*The carbon tax [29] serves as a widely cited Pigovian tax, making the implicit social costs of carbon emissions explicit by pricing them into market transactions.

- **Experiments in the Escape Room and challenging Cleanup environments** demonstrate the effectiveness of the proposed mechanism in alleviating social dilemmas in MARL.

## 2 Externality in MARL

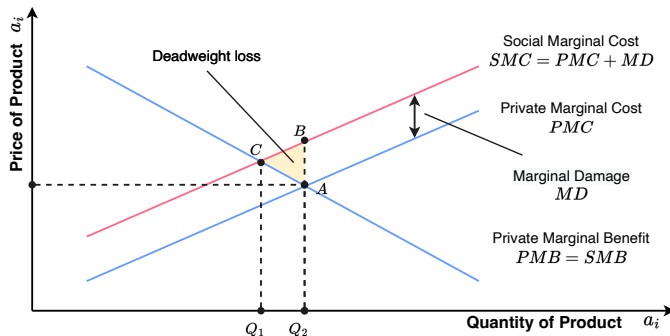

Figure 1: Externality [30]. The gap between social marginal cost and the private cost is externality.

This section illustrates the concept of externality in MARL and introduces a formalism for measuring it, enabling the visualization of social dilemmas. First, the concept of externality in economics is explained, with the graphical analysis [†] shown in Figure 1. Consider a firm $i$ that produces a product $a_i$ to meet consumer demand. Simultaneously, the firm generates pollution, which negatively impacts social welfare. Let us denote the quantity of produced $a_i$ as $q_i$. The price of $a_i$ is determined by a function that depends on both the quantity produced $q_i$ and the market demand for $a_i$. Defining the market demand as $q_i^r$, the profit function is represented as $P_i(q_i^r, q_i)$. The target of the firm is to maximize such utility:

$$u_i\left(q_i, q_i^r\right) = q_i \times P_i\left(q_i^r, q_i\right). \tag{1}$$

Let us analyze $N$ firms indexed $i = 1, 2, ..., N$, each firm $i$ producing a product $a_i$. Naturally, $i$ aims to maximize its profit by following (1). However, the production process inevitably generates activities not reflected in market transactions, such as pollution (which harms social welfare) or job creation (which benefits social welfare). To properly account for these externalities, social welfare assessments must incorporate these non-market activities. Therefore, we define the impact of such activities for each firm $i$ as a function $x_i(q_i)$ based on the quantity of product $a_i$ produced. Consequently, social welfare can be represented as:

$$U = \sum_i (u_i\left(q_i, q_i^r\right) + x_i(q_i)). \tag{2}$$

The externality is caused by these activities that are not reflected in market transactions, with the economic definition as:

**Definition 1.** *An **externality** occurs whenever one economic actor's activities affect another's activities in ways that are not reflected in market transactions [30].*

The influence $x_i$ can be used to measure the externality. When $x_i > 0$, it represents a positive externality. When $x_i < 0$, it represents a negative externality. We can express the Pigovian tax as a function $t_i(q_i)$ based on the quantity of product $a_i$ produced. The after-tax utility for firm $i$ is:

$$u_i\left(q_i, q_i^r\right) = q_i \times P_i\left(q_i^r, q_i\right) - t_i\left(q_i\right). \tag{3}$$

Here, the Pigovian tax $t_i(q_i)$ is designed to internalize the externality by making the firm's private cost align with the social cost, with the tax value directly proportional to the influence $x_i$. This ensures that addressing externalities is rooted in accurately quantifying the impact $x_i$ of an actor's activities on others. Similarly, in the context of multi-agent reinforcement learning (MARL), externalities can also emerge as a key concept, where agents' actions influence the outcomes or rewards of other agents

---

[†]Without considering social costs, the firm will seek to minimize its Private Marginal Costs at the expense of social welfare. By considering social costs, a firm can reduce the Social Marginal Cost, thereby promoting social welfare.

in ways that are not captured by their individual local rewards. By drawing an analogy between economic markets and MARL environments, an agent's action can be viewed as a form of market behavior, while its local reward corresponds to its individual payoff or utility. The externalities in this context then represent the unintended effects of an agent's actions on others, which aligns closely with the fundamental economic definition of externality. By extending this analogy, we can formalize the idea of externality in MARL as follows:

**Definition 2.** *An **externality** occurs whenever an agent's actions affect others in ways that are not reflected in individual local rewards.*

A decentralized MARL scenario is examined with an $N$-player partially observable general-sum Markov game on a finite set of states $\mathcal{S}$. At each timestep, each agent $i \in \{1, \ldots, N\}$ receives a $d$-dimensional observation $o_i \in \mathbb{R}^d$ from the observation function $\mathcal{O} : \mathcal{S} \times \{1, \ldots, N\} \to \mathbb{R}^d$, which maps the current environment state $s \in \mathcal{S}$ and agent identity to an individual observation. Based on its observation $o_i$, agent $i$ selects an action $a_i \in \mathcal{A}_i$ according to its policy $\pi_i(a_i \mid o_i)$, where $\mathcal{A}_i$ denotes the action space of agent $i$. which transitions to the next state $s'$ according to the transition function $P(s' \mid s, \mathbf{a})$ where $\mathbf{a} = (a_1, \ldots, a_N)$ denotes the joint action. Agents then receive their individual extrinsic rewards $r_i = \mathcal{R}_i(s, \mathbf{a})$. Each agent aims to maximize its long-term $\gamma$-discounted payoff:

$$Q^i(s, \mathbf{a}) = \mathrm{E}\left[\sum_{t=0}^{T} \gamma^t r_i(s^t, \mathbf{a}^t) \mid s^0 = s, \mathbf{a}^0 = \mathbf{a}\right]. \tag{4}$$

The social welfare of the scenario is defined as a global long-term $\gamma$-discount payoff as follows:

$$Q(s, \mathbf{a}, \mathbf{x}) = \mathrm{E}\left[\sum_{t=0}^{T} \gamma^t \sum_{i=1}^{N} (r_i(s^t, \mathbf{a}^t) + x_i(s^t, a_i^t)) \mid s^0 = s, \mathbf{a}^0 = \mathbf{a}\right], \tag{5}$$

where $x_i(s^t, a_i^t)$ represents the influence of agent $i$ on other agents in the scenario, and $\mathbf{x}$ denotes the joint influence $\{x_i\}_{i=1}^{N}$. In this setting, each agent's behavior inevitably affects the rewards of other agents. Consequently, social welfare is equivalent to:

$$Q(s, \mathbf{a}) = \mathrm{E}\left[\sum_{t=0}^{T} \gamma^t \sum_{i=1}^{N} r_i(s^t, \mathbf{a}^t) \mid s^0 = s, \mathbf{a}^0 = \mathbf{a}\right].$$

The optimal joint policy yields the following social welfare:

$$Q^*(s, \mathbf{a}^*) = \mathrm{E}\left[\sum_{t=0}^{T} \gamma^t \sum_{i=1}^{N} r_i(s^t, \mathbf{a}^t) \mid s^0 = s, \mathbf{a}^0 = \mathbf{a}^*\right],$$

where $\mathbf{a}^*$ represents the optimal joint action derived from the optimal joint policy. According to Definition 2, the externality of agent $i$ can be defined as follows:

$$E^i\left(s, \mathbf{a}_{-i}^*, a_i\right) = Q^*\left(s, \mathbf{a}^*\right) - Q\left(s, \mathbf{a}_{-i}^*, a_i\right), \tag{6}$$

where $\mathbf{a}_{-i}^*$ represents the joint optimal action excluding $a_i$, and $a_i$ is the current action of agent $i$. Based on (1), an Optimal Pigovian Tax reward shaping approach can be proposed to address externalities in MARL and resolve social dilemmas. The optimal Pigovian tax-based reward shaping can be expressed as:

$$F_i\left(s, \mathbf{a}_{-i}^*, a_i\right) = Q^*\left(s, \mathbf{a}^*\right) - Q\left(s, \mathbf{a}_{-i}^*, a_i\right). \tag{7}$$

The agent $i$ receives a modified reward with the reward shaping:

$$\hat{r}_i\left(s^t, \mathbf{a}^t\right) = r_i\left(s^t, \mathbf{a}^t\right) + F_i\left(s, \mathbf{a}{-i}^*, a_i\right), \tag{8}$$

which successfully internalizes the externality.

The Prisoner's Dilemma, a classic example of a social dilemma, is illustrated in Figure 2. In this scenario, two agents must independently choose between cooperation and defection. While the payoff matrix in Figure 2(a) shows that mutual cooperation maximizes collective welfare, defection remains the dominant strategy for each agent under self-interested reasoning. This misalignment between individual rationality and social welfare leads to an outcome with the lowest utility. The core of the Prisoner's Dilemma lies in the divergence between private incentives and social costs, which can be captured by the concept of externalities. Each agent neglects the negative externality their actions impose on the other. By quantifying these externalities through Equation 6 and applying Optimal Pigovian Tax reward shaping as defined in Equation 7, we transform the payoff structure into the revised matrix shown in Figure 2(b). In this modified matrix, the dominant strategy shifts to "Cooperate," demonstrating that by internalizing externalities via Optimal Pigovian Tax reward shaping, the social dilemma inherent in the Prisoner's Dilemma can be resolved.

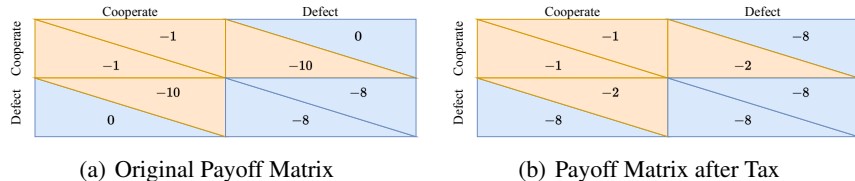

| (a) Original Payoff Matrix | (b) Payoff Matrix after Tax |

Figure 2: Pigovian Tax/Allowance for Prisoner's Dilemma.

# 3 Learning Optimal Pigovian Tax

In this section, LOPT will be explained in detail. As illustrated in Figure 3, it comprises two major components: **(1)** A centralized agent called *Tax Planner* that learns to allocate Pigovian tax and allowance rates by maximizing the long-term global rewards; **(2)** A reward shaping mechanism based on the learned tax/allowance allocation policy that internalizes each agent's externality, thereby aligning individual incentives with social welfare and effectively addressing social dilemmas. LOPT

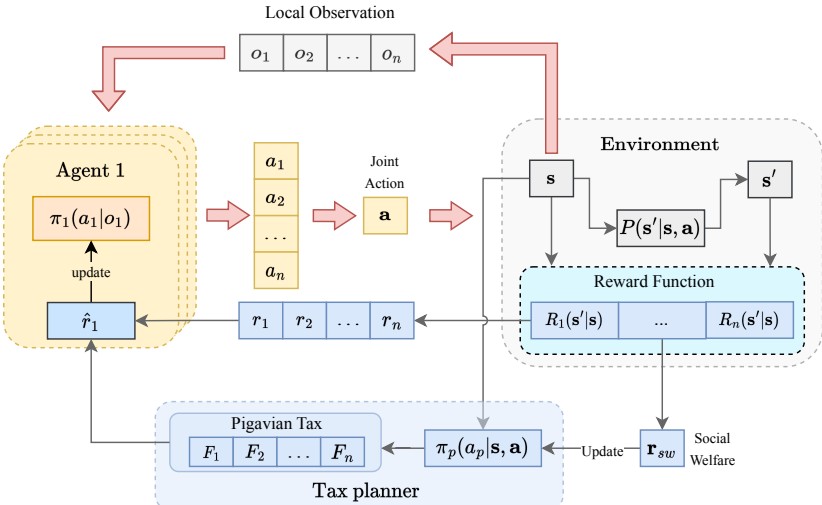

Figure 3: The Architecture of the **LOPT**. The centralized agent Tax planner allocate the Pigovian tax/allowance within a functional percentage formulation. Reward shaping is established based on the Pigovian tax/allowance to alleviate the social dilemmas.

is designed to learn the Optimal Pigovian Tax reward shaping described in (7) to internalize each agent's social cost. The Pigovian tax rewards is reformulated as:

$$F_*^i \left(s^t, \mathbf{a}_{-i}^t{}^*, a_i^t\right) = \sum_{j=0}^{N} r_j \left(s^t, \mathbf{a}^{t*}\right) - \sum_{j=0}^{N} r^j \left(s^t, \mathbf{a}_{-i}^t{}^*, a_i^t\right).$$

Pigovian tax reward shaping within percentage tax/allowance is formulated as:

$$F_{\boldsymbol{\theta},\boldsymbol{\delta}}^i \left(s^t, \mathbf{a}_{-i}^t{}^*, a_i^t\right) = -\theta_i r_i \left(s^t, \mathbf{a}_{-i}^t{}^*, a_i^t\right) + \delta_i(s^t, \mathbf{a}^t) \sum_{j=0}^{N} \theta_j r_j \left(s^t, \mathbf{a}_{-i}^t{}^*, a_i^t\right),$$

where $\boldsymbol{\theta}$ represents the tax rates for all agents, $\theta_i$ is the specific tax rate for agent $i$, while $\boldsymbol{\delta}$ denotes the allowance rates for all agents, and $\delta_i$ is the specific allowance rate for agent $i$. The Optimal Pigovian Tax reward shaping can be learned by determining appropriate values for $\boldsymbol{\theta}$ and $\boldsymbol{\delta}$, such that each $F_{\boldsymbol{\theta},\boldsymbol{\delta}}^i \left(s^t, \mathbf{a}-i^{t*}, a_i^t\right)$ equals $F^i * \left(s^t, \mathbf{a}_{-i}^t{}^*, a_i^t\right)$. However, since tax and allowance rates vary among different agents in different situations, it is necessary to represent $\boldsymbol{\theta}$ and $\boldsymbol{\delta}$ as functions of the current joint state and action. Therefore, the Pigovian tax reward shaping within percentage

tax/allowance is reformulated as:

$$F_{\boldsymbol{\theta},\boldsymbol{\delta}}^i\left(s^t, \mathbf{a}_{-i}^{t}{}^*, a_i^t\right) = -\theta_i(s^t, \mathbf{a}^t) r_i\left(s^t, \mathbf{a}_{-i}^{t}{}^*, a_i^t\right) + \delta_i(s^t, \mathbf{a}^t) \sum_{j=0}^N \theta_j(s^t, \mathbf{a}^t) r_j\left(s^t, \mathbf{a}_{-i}^{t}{}^*, a_i^t\right).$$

**Theorem 1.** *If other agents' actions are treated as part of the environment for any agent $i$ at any timestep $t$, there always exists typical $\theta_i(s^t, \mathbf{a}^t)$ and $\delta_i(s^t, \mathbf{a}^t)$ to let the $F_{\boldsymbol{\theta},\boldsymbol{\delta}}^i\left(s^t, \mathbf{a}_{-i}^{t}{}^*, a_i^t\right)$ equal to the $F_*^i\left(s^t, \mathbf{a}_{-i}^{t}{}^*, a_i^t\right)$.*[‡]

This theorem shows that the Pigovian tax reward shaping within percentage tax/allowance can reach the optimum in a specific condition. The theorem is proven in Appendix. B. The reward shaping function could be treated as follows:

$$F_{\boldsymbol{\theta},\boldsymbol{\delta}}^i\left(s^t, \mathbf{a}^t\right) = F_{\boldsymbol{\theta},\boldsymbol{\delta}}^i\left(s^t, \mathbf{a}_{-i}^{t}{}^*, a_i^t\right).$$

The central challenge is how to learn appropriate tax and allowance rate functions. As shown in Figure 3, we address this by introducing a centralized tax planner that treats tax and allowance rate as its action space and learns to maximize social welfare. The optimal Pigovian tax based on reward shaping is applied to internalize each agent's externality and solve the social dilemmas. In this form, the tax planner aims to learn the tax rates $\boldsymbol{\theta}$ and allowance rates $\boldsymbol{\delta}$ for all agents within the MARL task.

**Theorem 2.** *If the interactive influences from other agents are not considered, when the policy of tax planner $\langle \theta_i\left(s^t, \mathbf{a}^t\right), \delta_i\left(s^t, \mathbf{a}\right) \rangle$ maximizes the social welfare, the typical $F_{\boldsymbol{\theta},\boldsymbol{\delta}}^i\left(s^t, \mathbf{a}_{-i}^{t}{}^*, a_i^t\right)$ will qualitatively equivalent to the $F_*^i\left(s^t, \mathbf{a}_{-i}^{t}{}^*, a_i^t\right)$.*

Theorem 2 provides a key theoretical foundation for our approach, demonstrating that training the tax planner as a centralized reinforcement learning agent to maximize total social welfare implicitly approximates the optimal Pigovian tax. This theoretical equivalence is particularly significant, as it implies that **LOPT can explicitly quantify externalities in MARL by capturing social dilemmas** and internalize the broader societal impacts of self-interested agent behavior. In doing so, it directly addresses the core challenge of resolving social dilemmas in multi-agent reinforcement learning, as outlined in this paper. The complete proof is presented in Appendix B.

Guided by this insight, we formalize the tax planner as a reinforcement learning agent defined by the tuple $\langle \mathcal{S}_p, \mathcal{O}_p, \mathcal{A}_p, \mathcal{R}_p \rangle$, where at each timestep $t$: **(1).** The planner observes the global state and all agents' joint actions $o_p^t = \langle s^t, \mathbf{a}^t \rangle$; **(2).** selects taxes and allowances for agents $a_p^t = \langle \boldsymbol{\theta}^t, \boldsymbol{\delta}^t \rangle$; **(3).** receives a reward equal to the sum of all agents' rewards, $r_p^t$. Thus, the tax planner optimizes the cumulative social welfare:

$$\max_{\pi_p} J_p := \mathbb{E}\pi_p\left[\sum t = 0^T r_p(o_p^t, a_p^t)\right].$$

In short, by leveraging reinforcement learning to maximize social welfare, our method implicitly derives and implements optimal Pigovian tax-based reward shaping—**providing a principled and practical solution to accurately quantify and mitigate social dilemmas in MARL.**

In the training process, we use the approximated state-action function $Q_p(o_p, a_p)$ to replace the cumulative reward $r_p(o_p^t, a_p^t)$, and the objective function then becomes:

$$\max_{\pi_p} J_p := \mathbb{E}_{\pi_p}\left[Q\left(o_p, a_p\right)\right].$$

Typically, a policy gradient-based optimization [31] method is applied to train the tax planner. The gradient loss is therefore defined as follows:

$$\mathcal{L}(\phi_p) = \mathbb{E}_{\pi_p^{\phi_p}}\left[\nabla_{\pi_p^{\phi_p}} \log \pi_p\left(a_p^t \mid o_p^t\right) Q^{p,\pi_{\phi_p}^p}\left(o_p^t, a_p^t\right)\right],$$

where the tax planner's policy function parameters are represented by $\phi_p$. Additionally, to maintain balance between tax and allowance, the tax planner needs to minimize the following entropy $f(\pi_p)$ during the learning process:

$$f(\pi_p) = \left|\sum_{t=0}^T \sum_{i=0}^T F_{\boldsymbol{\theta},\boldsymbol{\delta}}^i\left(o^t, \mathbf{a}_{-i}^{t}{}^*, a_i^t\right)\right|,$$

---

[‡]Here we assume that the tax only occurs when the agent $i$ get an reward $r_i \neq 0$, because in reinforcement learning, its profit will only be shown in the step where $r \neq 0$.

As a result, the gradient loss $\mathcal{L}\left(\phi_p\right)$ can be denoted as:

$$\mathbb{E}_{\pi_p^{\phi_p}}\left[\nabla_{\pi_p^{\phi_p}} \log \pi_p\left(a_p^t \mid o_p^t\right) Q^{p, \pi_p^{\phi_p}}\left(o_p^t, a_p^t\right)\right] + \eta f\left(\pi_p^{\phi_p}\right), \tag{9}$$

where $\eta$ is a hyperparameter weighting the entropy $f(\pi_p)$.

In light of the learning process of the tax planner, other general agents are trained using the approximated Optimal Pigovian Tax reward shaping as follows:

$$\mathcal{L}\left(\phi_i\right) = \mathbb{E}_{\pi_i^{\phi_i}}\left[\nabla_{\pi_i^{\phi_i}} \log \pi^i\left(a_i \mid s\right) \hat{Q}^{i, \pi_i^{\phi_i}}(s, \mathbf{a})\right], \tag{10}$$

where function $\hat{Q}^{i, \pi_i^{\phi_i}}(s, \mathbf{a})$ is defined as

$$r_i(s, \mathbf{a}) + F^i\left(s, \mathbf{a}^{-i^*}, a_i\right) + \gamma \max_{\mathbf{a}'} \hat{Q}^{i, \pi_i^{\phi_i}}\left(s', \mathbf{a}'\right).$$

The typical learning process of LOPT is outlined in Algorithm 1 (Appendix), and its performance is demonstrated through experiments in the Escape Room and Cleanup environments.

## 4 Experiment

**Environments** We conduct experiments on both the ESCAPE ROOM [50] and the CLEANUP [15] environments, the details are summarized as follows:

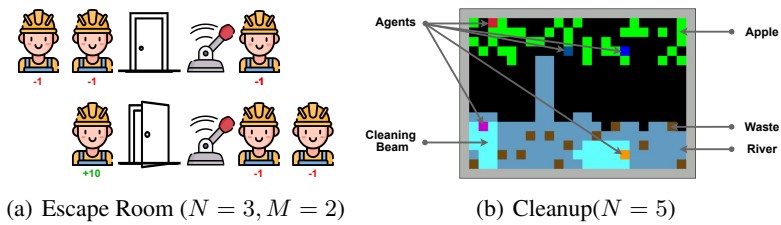

(a) Escape Room ($N = 3, M = 2$)      (b) Cleanup($N = 5$)

Figure 4: Environment Examples

*Escape Room (ER)*: In an Escape Room game ER($N$, $M$), where $N > M$, $N$ agents as players aim to escape from the room (Figure 4(a)). In this environment, there are 3 available states: *door*, *lever*, and *start* (the initial state), where agents are able to take actions to keep or change their states. An agent is able to open the *door*, then receive an extrinsic reward of $+10$, and end the current episode if and only if no less than $M$ other agents pull the *lever*. Otherwise, agents will receive an extrinsic penalty of $-1$ for making any state change. When agents try to maximize their rewards egoistically, they tend to stay in current positions to avoid punishments or move to the *door* and wait for others to pull the *lever* that will never happen, which creates a social dilemma. In our experiments, settings of ($N = 2, M = 1$) and ($N = 3, M = 2$) are applied.

*Cleanup*: In a Cleanup game with $N$ agents (Figure 4(b)), agents get an extrinsic reward of $+1$ by harvesting an apple and aim to collect as many apples as possible. Apples are spawned at a variable rate, which decreases linearly as the aquifer fills with waste over time. If the waste density reaches the depletion threshold, no more apples will spawn, so agents must clean waste without any extrinsic reward, creating a social dilemma. At each timestep $t$, agents observe their surroundings as an image and perform one of the following actions:

$$\left\{\begin{array}{c} \text{move left, move right, move up, move down, stay,} \\ \text{rotate clockwise, rotate counterclockwise, fire cleaning beam} \end{array}\right\},$$

where the *move"* / *rotate"* actions change the positions/directions of agents in the map, the *stay"* action waits at the original positions and does nothing, and the *fire cleaning beam"* action allows agents to fire cleaning beams (with width 3) to clean wastes (the beam cannot penetrate wastes). To verify how the proposed **LOPT** resolves the social dilemma, we initialize each episode with sufficient wastes and no spawned apple, then experiment with $N = 2$ on a $7 \times 7$ map and a $10 \times 10$ map, where the latter applies lower depletion threshold and apple respawn rate. Finally, a more complex scenario of $N = 5$ Cleanup games with a larger $18 \times 25$ map and a much lower apple respawn rate is used to explore the generalizability and scalability of our proposed method.

**Implementation and Baselines**  We compared several baseline approaches in our experiments. First, we evaluated standard reinforcement learning algorithms including Policy Gradient (**PG**) for Escape Room, and Actor-Critic (**AC**) along with Proximal Policy Optimization (**PPO**) for Cleanup.

We then examined state-of-the-art methods for addressing social dilemmas: **LIO** [50] and its decentralized variant **LIO-dec**, which learn to incentivize cooperation through reward-sharing; Inequity Averse (**IA**) [15], which promotes cooperation via inequity-averse social preferences; Model of Other Agents (**MOA**) [17], which uses counterfactual reasoning to model agent interactions; and Social Curiosity Module (**SCM**) [14], which combines curiosity and empowerment rewards.

For specific environments, we implemented various method combinations. In Escape Room, we compared **LIO**, **LIO-dec**, and Policy Gradient variants with discrete and continuous reward-giving actions (**PG-d/c**). The Cleanup($N = 2$) evaluation included **LIO**, **IA**, **MOA**, **SCM**, and Actor-Critic variants (**AC-d/c**), while the more complex Cleanup($N = 5$) scenario focused on **MOA** and **SCM**.

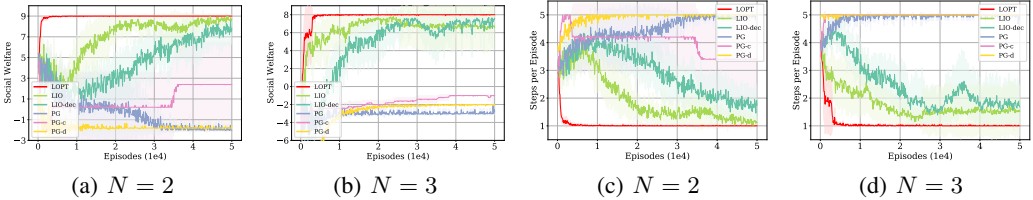

(a) $N = 2$  (b) $N = 3$  (c) $N = 2$  (d) $N = 3$

Figure 5: Results on Escape Room Environment. (5(a), 5(b)) shows the learning curves of the proposed **LOPT**; which converges to the optimum and successfully solves the Escape Room social dilemmas. (5(c), 5(d)) shows **LOPT** is able to end the episode in a single 1 step without any betrayal.

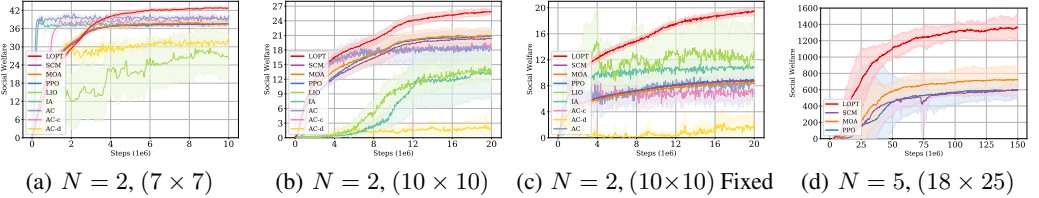

(a) $N = 2, (7 \times 7)$  (b) $N = 2, (10 \times 10)$  (c) $N = 2, (10 \times 10)$ Fixed  (d) $N = 5, (18 \times 25)$

Figure 6: Results on Cleanup Environment. (6(a), 6(b)) shows the learning curves for the proposed **LOPT** in Cleanup($N = 2$); (6(c)) shows the learning curves for the proposed **LOPT** in Cleanup($N = 2$) with the fixed-orientated assumption. (6(d)) scales to a more complex environment with $N = 5$.

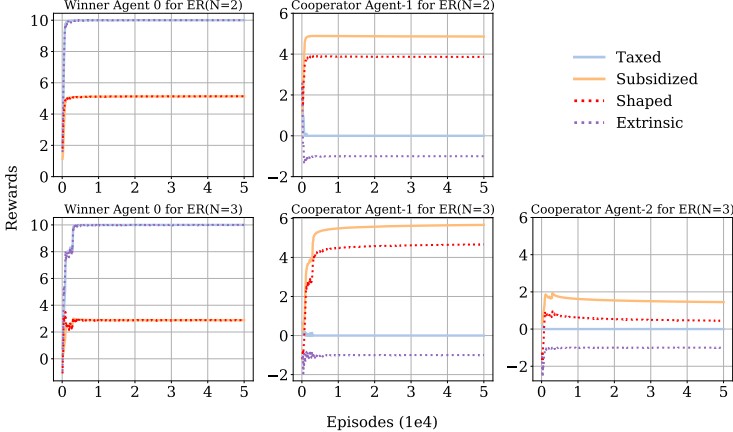

Figure 7: Rewards for Each Agent with Different Behaviors in Escape Room Environment. **LOPT** internalizes externalities and redistributes rewards among agents with taxes and allowances.

**Results**  Our experiments demonstrate that the proposed **LOPT** successfully resolves social dilemmas by approximating externalities among agents in MARL problems and modeling the optimal

Pigovian tax reward shaping. This approach internalizes the externalities, enabling convergence toward optimal solutions even in complex scenarios. In both Escape Room and Cleanup environments, **LOPT** implements effective tax/allowance schemes and redistributes rewards among agents, thereby internalizing externalities and guiding agents to develop social-good behaviors (both cooperative and competitive), which significantly accelerates learning curves. Additionally, compared to baseline methods, the internalized externalities in our proposed **LOPT** result in fewer betrayals, leading to a more stable learning process.

*Escape Room*. In both ER($N = 2, M = 1$) and ER($N = 3, M = 2$) settings, Figures 5(a) and 5(b) demonstrate that **LOPT** rapidly converges to optimal values (8 and 9 respectively) by leveraging optimal Pigovian tax incentives. **PG** agents completely fail due to selfish optimization, while **PC-d/c** agents exhibit high variance and suboptimal performance. Although **LIO** and **LIO-dec** achieve near-optimal results, they display instability and betrayal-related fluctuations are absent to **LOPT**. The optimal solution requires only 1 step ($M$ agents pull levers, $N - M$ open door). Figures 5(c) and 5(d) confirm that **LOPT** consistently achieves this efficiency, unlike other methods. Figure 7 reveals the underlying mechanism: **LOPT** taxes "Winner" agents (those creating negative externalities) and rewards "Cooperator" agents (those generating positive externalities), effectively internalizing externalities through Pigovian incentives.

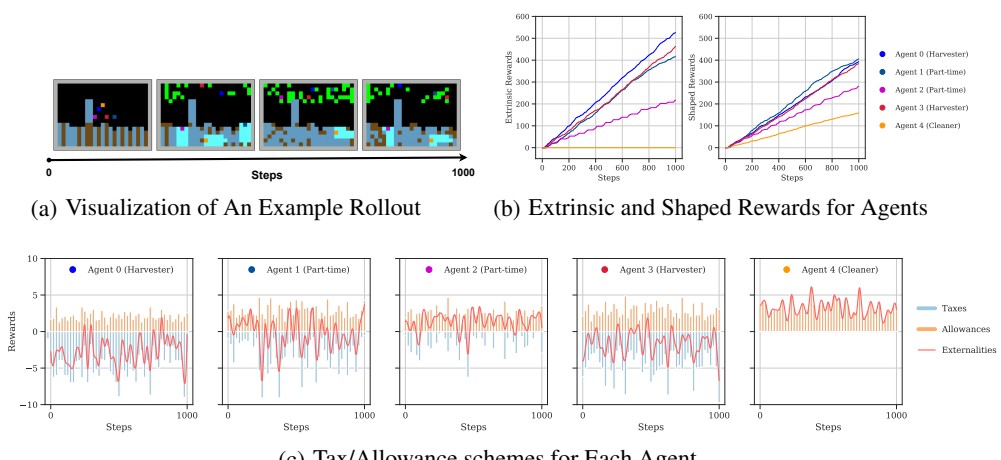

(a) Visualization of An Example Rollout    (b) Extrinsic and Shaped Rewards for Agents

(c) Tax/Allowance schemes for Each Agent

Figure 8: An Example Rollout for Cleanup($N = 5$) Environment. (8(a)) visualizes this example rollout, where agents apply different social-good behaviors and divisions of laborers (cleaner, harvester, and part-time) emerge. (8(b)) shows the approximated optimal Pigovian tax reward shaping by the proposed **LOPT**. (8(c)) shows the reward shaping process of the **LOPT** in this episode, which demonstrates how the **LOPT** internalizes externalities for agents with different socially contributed behaviors.

*Cleanup*. We evaluate **LOPT** on Cleanup with both simple ($N = 2$) and complex ($N = 5$) scenarios. For $N = 2$, we remove LIO's rotation-action restriction, testing on $7 \times 7$ and $10 \times 10$ maps. Figures 6(a) and 6(b) show **LOPT** achieves near-optimal social welfare, while **LIO** fails to learn efficient policies. **AC-d** performs well on $7 \times 7$ but poorly scales to $10 \times 10$. Other baselines reach near-optimum on $7 \times 7$, but **IA** and **AC-c** degrade severely on $10 \times 10$ compared to **AC**, **PPO**, **SCM**, and **MOA**. Even with fixed-orientation (Figure 6(c)), **LOPT** maintains stable performance by properly internalizing externalities, while **LIO** shows instability due to potential incentive misalignment. We then compare the proposed **LOPT** with **PPO**, **SCM**, and **MOA** baselines, which have shown better scalability, in the more complex Cleanup($N = 5$) scenario, where an $18 \times 25$ large map and applied apple respawn rate are applied. Figure 6(d) shows that our proposed **LOPT** is able to scale to more complex scenarios and internalize the approximated externalities by learning optimal Pigovian tax reward shaping, which effectively helps agents to learn in social dilemmas. To demonstrate how **LOPT** estimates externalities and influences agent behaviors, we analyze their actions and reward redistribution. Figure 8(a) shows a Cleanup game with $N = 5$ agents: Initially, agents 1, 2, and 4 clean waste (exceeding the depletion threshold) to accelerate apple spawning. Agent 4 becomes a full-time cleaner while agent 1 transitions to part-time harvesting. Agent 2 becomes another part-timer, balancing harvesting with waste cleaning, while agents 0 and

3 remain full-time harvesters. **LOPT** naturally induces labor specialization (cleaners, harvesters, and part-timers) by internalizing externalities, effectively addressing the social dilemma. Figure 8(c) reveals the mechanism: Harvesters (0, 3) pay heavy taxes for negative externalities; part-timers (1, 2) receive allowances for cleaning but pay taxes for harvesting; cleaner 4 gains substantial allowances for positive externalities. The system provides near-optimal Pigovian tax incentives (Figure 8(b)) to guide agents toward superior outcomes. Additional results appear in Appendix D.3.

## 5 Conclusion

In this paper, we introduce externality theory to measure the influence of agents' behavior on social welfare. Based on this theoretical foundation in the MARL domain, we propose the **L**earning **O**ptimal **P**igovian **T**ax method to address social dilemmas. We construct a centralized agent, Tax Planner, which learns the tax/allowance allocation policy for each agent. Through Optimal Pigovian Tax reward shaping, each agent's externality is internalized, encouraging behaviors that benefit social welfare. Our experiments demonstrate the superiority of the proposed mechanism in alleviating social dilemmas in MARL. For future work, we aim to develop a decentralized Pigovian tax/allowance mechanism to learn reward shaping that internalizes agents' externalities while reducing computational complexity.

## 6 Acknowledge

Xiangfeng Wang is supported by the National Key R&D Program of China (Nos. 2021YFA1000300 and 2021YFA1000302), the NSFC (62231019) and SHEITC (2025-GZL-RGZN-BTBX-01004). Wenhao Li is supported by the NSFC (62406270) and the STCSM Shanghai Rising-Star Program (24YF2748800). Jun Luo is supported by the NSFC (72031006).

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

# A  Related Work

Our work, LOPT, is motivated by the challenge of fostering cooperation among independently learning agents in *intertemporal social dilemmas (ISDs)* [22]. In ISDs, agents pursue individual long-term returns, but mutual defection often leads to suboptimal collective outcomes and degraded social welfare over time.

## A.1  Limitations of Conventional MARL in ISDs

Conventional Multi-Agent Reinforcement Learning (MARL) algorithms designed for *fully cooperative tasks* [13, 41, 38, 24, 25, 10] struggle with ISDs due to their assumption of aligned agent incentives. In contrast, ISDs feature *mixed motivations*, where agents' local optima may conflict with collective well-being.

Several approaches attempt to address this by incorporating *reward shaping* or *intrinsic motivation* [12, 15, 47]. However, these methods often rely on hand-crafted heuristics or evolution-based adaptations to other agents' behaviors, limiting generality and scalability. More recent approaches, like LIO [50], enable agents to learn incentives for others, while some studies explore *mechanism or information design* [27] in fully cooperative contexts. Yet, these methods typically lack a unified economic rationale for shaping rewards.

## A.2  Externality Theory and Economic Inspiration

LOPT is grounded in *externality theory* [30], which provides a principled framework for aligning individual incentives with social welfare—a central challenge in ISDs. In both *non-market* [1] and *market economies* [35], various mechanisms have been developed to internalize externalities, such as the *Pigovian tax* [5], which penalizes behaviors that impose social costs.

Our approach adopts a learning-based Pigovian tax framework to shape agent incentives and mitigate negative externalities. This aligns with economic findings that reward structures significantly influence cooperative behavior in repeated settings. For instance, [39] demonstrates that limited feedback and longer interaction horizons promote cooperation in human queueing systems, emphasizing the role of information and interaction design. Similarly, [4] shows that optimal mechanisms in competitive markets are sensitive to network structures, reinforcing the importance of structural design in multi-agent coordination.

Moreover, [33] highlights the theoretical interchangeability of taxes and subsidies under certain conditions, broadening the space of policy tools for influencing agent behavior. While LOPT focuses on tax-based shaping, its theoretical foundation can naturally extend to subsidy schemes depending on fairness or implementation considerations.

Our design also draws structural inspiration from the AI Economist [51], employing a two-stage architecture to learn tax policies. However, LOPT specifically targets ISDs in MARL and distinguishes itself by leveraging externality theory to inform its reward shaping paradigm.

## A.3  Structural Solutions to ISDs: Centralized vs. Decentralized

Beyond reward shaping, recent work has explored *structural interventions* for ISDs, drawing parallels to economic governance models. These can be categorized into:

- **Centralized boundaries** [9, 16], which emulate government-like authorities to regulate agent behavior.
- **Decentralized sanctions** [3, 20, 48, 45, 11], which enable agents to punish others for socially harmful behavior.

LOPT follows the **centralized boundaries** paradigm, introducing a centralized tax planner that learns to enforce Pigovian taxes based on global observations. Unlike previous centralized approaches, such as [9], which uses arbitrary allocation for shared resources, or [16], which introduces a fixed tax mechanism, LOPT *learns a dynamic tax policy* tailored to the environment. Furthermore, our method is *theoretically supported by externality theory*, providing a principled foundation for shaping agent behavior.

## B Proof

**Theorem 1.** *If other agents' actions are treated as part of the environment for any agent $i$ at any timestep $t$, there always exists typical $\theta_i(s^t, \mathbf{a}^t)$ and $\delta_i(s^t, \mathbf{a}^t)$ to let the $F^i_{\theta,\delta}\left(s^t, \mathbf{a}^{t\,*}_{-i}, a^t_i\right)$ equal to the $F^i_*\left(s^t, \mathbf{a}^{t\,*}_{-i}, a^t_i\right)$.*

*Proof.* We make classified discussions for any agent $i$ create negative externality, agent $i$ create positive externality. For any agent $i$ which creates a negative externality at timestep $t$: the agent will not receive any allowance, so the allowance rate function $\delta_i(s^t, a^t_i)$ is equal to 0. And the tax rate can be written as:

$$\theta_i(s^t, a^t_i, a^{t\,*}_{-i}) = \frac{E^i(s^t, a^{t\,*}_{-i}, a^t_i)}{r_i(s^t, a^t_i, a^{t\,*}_{-i})}, \tag{11}$$

$$\theta_i(s^t, a^t_i, a^{t\,*}_{-i}) = \frac{Q(s^t, \mathbf{a}^{t*}) - Q(s^t, a^{t\,*}_{-i}, a^t_i)}{r_i(s^t, a^t_i, a^{t\,*}_{-i})} \tag{12}$$

And as the interactive influence from other agents is not considered, other agents' optimal action $a^{t\,*}_{-i}$ can be seen as a part of the environment, and this optimum has a fixed result. Therefore, like the reinforcement learning method with an advantage function, for each agent $i$, the advantage function based on the current joint state and action can also be found in the tax rate, where:

$$Q(s^t, \mathbf{a}^{t*}) = A^0_i(s^t, \mathbf{a}^t) \times Q(s^t, \mathbf{a}^t),$$
$$Q(s^t, a^{t\,*}_{-i}, a^t_i) = A^1_i(s^t, \mathbf{a}^t) \times Q(s^t, \mathbf{a}^t), \tag{13}$$
$$r_i(s^t, a^t_i, a^{t\,*}_{-i}) = A^2_i(s^t, \mathbf{a}^t) \times r_i(s^t, \mathbf{a}^t).$$

Then the tax rate for agent $i$ becomes:

$$\theta_i(s^t, a^t_i, a^{t\,*}_{-i}) = \frac{(A^0_i(s^t, \mathbf{a}^t) - A^1_i(s^t, \mathbf{a}^t)) \times Q(s^t, \mathbf{a}^t)}{A^2_i(s^t, \mathbf{a}^t) \times r_i(s^t, \mathbf{a}^t)},$$
$$\theta_i(s^t, \mathbf{a}^t) = \frac{(A^0_i(s^t, \mathbf{a}^t) - A^1_i(s^t, \mathbf{a}^t)) \times Q(s^t, \mathbf{a}^t)}{A^2_i(s^t, \mathbf{a}^t) \times r_i(s^t, \mathbf{a}^t)}. \tag{14}$$

Then it is proven that for any agent $i$ which generates negative externality, there always exists typical $\theta_i(s^t, \mathbf{a}^t)$ and $\delta_i(s, \mathbf{a}^t)$ to let the $F^i_{\theta,\delta}\left(s^t, \mathbf{a}^{t\,*}_{-i}, a^t_i\right)$ equivalent to the $F^i_*\left(s^t, \mathbf{a}^{t\,*}_{-i}, a^t_i\right)$.

Similarly, for any agent $i$ which generates positive externality, there also exists typical $\theta_i(s^t, \mathbf{a}^t)$ and $\delta_i(s^t, \mathbf{a}^t)$ to satisfy the condition above.

This proves that if the interactive influence from other agents is not considered, for any agent $i$ at any timestep $t$, there always exists typical $\theta_i(s^t, \mathbf{a}^t)$ and $\delta_i(s, \mathbf{a}^t)$ to let the $F^i_{\theta,\delta}\left(s^t, \mathbf{a}^{t\,*}_{-i}, a^t_i\right)$ equivalent to the $F^i_*\left(s^t, \mathbf{a}^{t\,*}_{-i}, a^t_i\right)$. $\square$

**Theorem 2.** *If the interactive influences from other agents are not considered, when the policy of tax planner $\langle\theta_i\left(s^t, \mathbf{a}^t\right), \delta_i\left(s^t, \mathbf{a}\right)\rangle$ maximizes the social welfare, the typical $F^i_{\theta,\delta}\left(s^t, \mathbf{a}^{t\,*}_{-i}, a^t_i\right)$ will qualitatively equivalent to the $F^i_*\left(s^t, \mathbf{a}^{t\,*}_{-i}, a^t_i\right)$.*

*Proof.* Here we use the method of "reduction to absurdity." Suppose that there exists an agent $i$ which generates negative externality, and the learned $F^i_{\theta,\delta}\left(s^t, \mathbf{a}^{t\,*}_{-i}, a^t_i\right)$ does not qualitatively equivalent to the $F^i_*\left(s^t, \mathbf{a}^{t\,*}_{-i}, a^t_i\right)$. The reason why agent $i$ will choose the selfish behavior which harms social welfare without reward shaping is because its individual reward shows:

$$r_i(s^t, a^{t\,*}_{-i}, a^t_i) > r_i(s^t, \mathbf{a}^{t*}). \tag{15}$$

And the effect of the Optimal Pigovian Tax reward shaping is to let any $a^t_i \in A_i$ hold the following constraint:

$$r_i(s^t, a^{t\,*}_{-i}, a^t_i) + F^i_{\theta,\delta}(s^t, a^{t\,*}_{-i}, a^t_i) < r_i(s^t, \mathbf{a}^{t*}). \tag{16}$$

As we suppose that its typically learned reward shaping does not qualitatively equivalent to the Optimal Pigovian Tax reward shaping. That means there exists some $a^t_i \in A_i$, which causes:

$$r_i(s^t, a^{t\,*}_{-i}, a^t_i) + F^i_{\theta,\delta}(s^t, a^{t\,*}_{-i}, a^t_i) > r_i(s^t, \mathbf{a}^{t*}). \tag{17}$$

This means agent $i$ within its optimal policy $\pi_i^*$ would like to choose the behavior $a_i^t$ rather than the behavior in optimal joint actions $\mathbf{a}^{t\,*}$. Then if we use the tax planner's learned policy $\pi_p^{\phi_p}$ to describe the tax rate allocation, which means there exists another tax planner's policy $\pi_p^*$, letting:

$$\mathbb{E}_{\pi_p^{\phi_p}}\left[\sum_{t=0}^{T} r_p\left(s_p^t, a_p^t\right)\right] < \mathbb{E}_{\pi_p^*}\left[\sum_{t=0}^{T} r_p\left(s_p^t, a_p^t\right)\right]. \tag{18}$$

Thus we have shown that if any learned reward shaping of agent $i$ is not qualitatively equivalent to the Optimal Pigovian Tax reward shaping, the tax planner's learned policy is not optimal. $\qquad\square$

## C  Algorithm

---
**Algorithm 1 LOPT**: Learning Optimal Pigovian Tax
---
1: Initialization: all general agents' policy parameters $\{\phi_i\}$, tax planner's policy parameters $\phi_p$;
2: **for** each iteration **do**
3:     Generate a joint state-action trajectory with shaped rewards and tax/allowance rates as $\{\tau\}$;
4:     **for** each state-action pair with shaped reward for each agent $i$, i.e., $\langle s_i, \mathbf{a}, r_i + F_i \rangle$ in $\{\tau\}$ **do**
5:         Compute the new $\hat{\phi}_i$ by gradient ascent on (10);
6:     **end for**
7:     **for** each tax planner state-action pair with global reward $\langle o_p, a_p, r_p \rangle$ in $\{\tau\}$ **do**
8:         Compute the new $\hat{\phi}_p$ by gradient ascent on (9);
9:     **end for**
10:    $\phi_i \leftarrow \hat{\phi}_i$, $\phi_p \leftarrow \hat{\phi}_p$, for all $i \in \mathbb{N}$.
11: **end for**
---

# D   Experiment

## D.1   Implementations

The policy and value functions in LOPT are implemented as neural networks (detailed architecture provided in Appendix. D.2). Training is conducted on a virtual machine hosted on a GPU server equipped with four NVIDIA GTX 2080 Ti GPUs, a 24-core CPU, and 32 GB of DRAM.

We implemented the **LOPT** in both Escape Room and Cleanup environments. At each timestep $t$, the global observation $o_{global}^t$ from the joint state $s_t$, and the joint action $\mathbf{a}^t$ are fed to the tax planner as input. To better handle our challenging environments, we provide a "bank" variable to the tax planner to save rewards from taxes as available budgets for allowances, which supports the more sophisticated tax/allowance mechanism. Then, the current bank state $o_{bank}^t$ and joint reward $\mathbf{r}^t$ are also introduced to the observation:

$$o_p^t = \left\langle o_{global}^t, \mathbf{a}^t, o_{bank}^t, \mathbf{r}^t \right\rangle.$$

The tax planner outputs the joint tax rate $\boldsymbol{\theta}^t$ and the joint allowance rate $\boldsymbol{\delta}^t$. In addition, the tax planner outputs. Also, it outputs a percentage for rewards withdrawn from the bank as the budget ratio $a_t^{bank}$. So, the action for the current time step is:

$$a_t^p = \left\langle \boldsymbol{\theta}^t, \boldsymbol{\delta}^t, a_{bank}^t \right\rangle.$$

In addition, the entropy $f(\pi_p)$ is weighted by a hyperparameter $\eta$ in (9) Concretely, in both environments with $N$ agents, $o_{bank}^t$ and $\mathbf{a}^t$ are scalers, while $\mathbf{a}^t$, $\mathbf{r}^t$, $\boldsymbol{\theta}^t$ and $\boldsymbol{\delta}^t$ are $N$ dimensional vectors. In the Escape Room games, the tax planner agent observes a multi-hot vector global states $o_{global}^t \in \{0,1\}^d$ from the joint state $s_t$, where $d = 3N$. And in the Cleanup games, the global observation $o_{global}^t$ is the global visual normalized RGB observation with the same width and height of the applied map.

In the Escape Room environment, the policy network for the tax planner is defined as follows: 1). a dense layer $h1_1$ of size 64 takes $o_{global}^t$ as input and 3 dense layers $h1_i, i = 2, 3, 4$ of size 32 for $\mathbf{a}^t$, $o_{bank}^t$, and $\mathbf{r}^t$ respectively; 2). the outputs of dense layers $h1_i, i = 1, 2, 3, 4$ are concatenated and fed to a dense layer $h2$ of size 32; 3). the output of dense layer $h2$ is fed to 3 dense layers $h3_i, i = 1, 2, 3$ of sizes 1, $N$, $N$ and activation functions $sigmoid$, $sigmoid$, $softmax$, then output as $a_t^{bank}$, $\boldsymbol{\theta}^t$, $\boldsymbol{\delta}^t$ respectively. While in the Cleanup environment, the policy network for the tax planner is defined as follows: 1). the global observation $o_{global}^t$ is firstly fed to a convolutional layer $conv1$ of kernel size $3 \times 3$, stride 1 and 6 filters; 2). the output of the convolutional layer $conv1$, $\mathbf{a}^t$, $o_{bank}^t$, and $\mathbf{r}^t$ are fed to 4 two-layer dense layers $h2_i, i = 1, 2, 3, 4$ of size 32 and 32 respectively; 3). the outputs of dense layers $h2_i, i = 1, 2, 3, 4$ are concatenated and fed to an LSTM of cell size 128; 4). at last, the output of the LSTM is fed to the dense layers and output as $a_t^{bank}$, $\boldsymbol{\theta}^t$, $\boldsymbol{\delta}^t$ respectively.

The settings of hyperparameters for baselines follow their previous work [15, 17, 50, 14]. For all experiments, the tuned hyperparameters of all baselines and LOPT are given in Table. 2-4 in the appendix D.2, where: $\alpha$ is the learning rate; $\alpha_{schedule}$ is a list that contains the step and weight pairs for the learning rate scheduler; $\eta$ is the weight for the entropy $f(\pi_p)$; $\epsilon$ in [50] decays linearly from $\epsilon_{start}$ to $\epsilon_{end}$ by $\epsilon_{div}$ episodes; $\beta$ is coefficient for the entropy of the policy.

## D.2 Hyperparameter

| Parameters | $N = 2$, $7 \times 7$ map | $N = 2$, $10 \times 10$ map | $N = 2, 10 \times 10$ map fixed orientations | $N = 5$, $18 \times 25$ map |
|---|---|---|---|---|
| appleRespawnProbability | 0.5 | 0.3 | 0.3 | 0.05 |
| wasteSpawnProbability | 0.5 | 0.5 | 0.5 | 0.5 |
| thresholdDepletion | 0.6 | 0.4 | 0.4 | 0.4 |
| thresholdRestoration | 0.0 | 0.0 | 0.0 | 0.0 |
| rotationEnabled | ✓ | ✓ | ✗ | ✓ |
| view_size | 4 | 7 | 7 | 7 |
| max_steps | 50 | 50 | 50 | 1000 |

Table 1: Experiment Settings for Cleanup Environment.

| Hyperparameters | $N = 2$ | | | | | | $N = 3$ | | | | | |
|---|---|---|---|---|---|---|---|---|---|---|---|---|
| | PG | PG-d | PG-c | LIO | LIO-dec | LOPT | PG | PG-d | PG-c | LIO | LIO-dec | LOPT |
| $\alpha$ | 1e−4 | 1e−4 | 1e−3 | 1e−4 | 1e−4 | 1e−3 | 1e−4 | 1e−4 | 1e−3 | 1e−4 | 1e−4 | 1e−3 |
| $\eta$ | - | - | - | - | - | 0.95 | - | - | - | - | - | 0.95 |
| $\epsilon_{\text{start}}$ | 0.5 | 0.5 | 1.0 | 0.5 | 0.5 | 0.5 | 0.5 | 0.5 | 1.0 | 0.5 | 0.5 | 0.5 |
| $\epsilon_{\text{end}}$ | 0.05 | 0.05 | 0.1 | 0.1 | 0.1 | 0.05 | 0.05 | 0.05 | 0.1 | 0.3 | 0.3 | 0.05 |
| $\epsilon_{\text{div}}$ | 100 | 100 | 1000 | 1000 | 1000 | 100 | 100 | 100 | 1000 | 1000 | 1000 | 100 |
| $\beta$ | 0.01 | 0.01 | 0.1 | 0.01 | 0.01 | 0.01 | 0.01 | 0.01 | 0.1 | 0.01 | 0.01 | 0.01 |

Table 2: Hyperparameter Settings for Escape Room Environment.

| Hyperparameters | $7 \times 7$ map | | | | | | | | | $10 \times 10$ map | | | | | | | | |
|---|---|---|---|---|---|---|---|---|---|---|---|---|---|---|---|---|---|---|
| | AC | AC-d | AC-c | IA | LIO | PPO | MOA | SCM | LOPT | AC | AC-d | AC-c | IA | LIO | PPO | MOA | SCM | LOPT |
| $\alpha$ | 1e−3 | 1e−4 | 1e−3 | 1e−3 | 1e−4 | 2.52e−3 | 2.52e−3 | 2.52e−3 | 2.52e−3 | 1e−3 | 1e−3 | 1e−3 | 1e−3 | 1e−4 | 1.26e−3 | 1.26e−3 | 1.26e−3 | 2.52e−3 |
| $\alpha_{schedule}$ | - | - | - | - | - | [ (5e5, 1.26e−3), (2.5e6, 1.26e−4) ] | | | | - | - | - | - | - | [ (1e7, 1.26e−4) ] | | | [(5e5, 1.26e−3), (1e7, 1.26e−4)] |
| $\eta$ | - | - | - | - | - | - | - | - | 0.95 | - | - | - | - | - | - | - | - | 0.95 |
| $\epsilon_{\text{start}}$ | 0.5 | 0.5 | 0.5 | 0.5 | 0.5 | - | - | - | - | 0.5 | 0.5 | 1.0 | 0.5 | 0.5 | - | - | - | - |
| $\epsilon_{\text{end}}$ | 0.05 | 0.05 | 0.05 | 0.05 | 0.05 | - | - | - | - | 0.05 | 0.05 | 0.05 | 0.05 | 0.05 | - | - | - | - |
| $\epsilon_{\text{div}}$ | 100 | 100 | 100 | 1000 | 100 | - | - | - | - | 5000 | 1000 | 1000 | 5000 | 1000 | - | - | - | - |
| $\beta$ | 0.1 | 0.1 | 0.1 | 0.1 | 0.1 | 1.76e−3 | 1.76e−3 | 1.76e−3 | 1.76e−3 | 0.01 | 0.01 | 0.1 | 0.01 | 0.01 | 1.76e−3 | 1.76e−3 | 1.76e−3 | 1.76e−3 |

Table 3: Hyperparameter Settings for Cleanup($N = 2$) Environment.

| Hyperparameters | PPO | MOA | SCM | LOPT |
|---|---|---|---|---|
| $\alpha$ | 1.26e−3 | 1.26e−3 | 1.26e−3 | 1.26e−3 |
| $\alpha_{schedule}$ | [ (2e7, 1.26e−4), (2e8, 1.26e−5) ] | | | [(2.5e7, 1.26e−4)] |
| $\eta$ | - | - | - | 0.95 |
| $\beta$ | 1.76e−3 | 1.76e−3 | 1.76e−3 | 1.76e−3 |

Table 4: Hyperparameter settings for Cleanup($N = 5$).

## D.3 Addtional Experiment Results

In this section, additional results from experiments will be demonstrated. As illustrated in Figure. 9-12, our proposed **LOPT** is able to internalize externalities in all of our Cleanup experiment settings and provide approximated optimal Pigovian tax reward shaping to greatly alleviate the social dilemmas. And for Cleanup($N = 5$) environment, we further show the relationship among the environmental states of the numbers of apples and wastes and the tax/allowance schemes given by the **LOPT**, where proper tax/allowance schemes are given for agents with different socially contributed behaviors in Figure 13(a), Figure 13(b), and Figure 13(c) Also, Figure. 13(d) shows that the **LOPT** encourages agents to clean wastes efficiently and maintains the density of wastes at a relatively low level so that the apples are spawned at a relatively high rate. Also, we provide visualized and analyzed results from example rollouts in Cleanup($N = 2$) with both the $7 \times 7$ and the $10 \times 10$ maps.

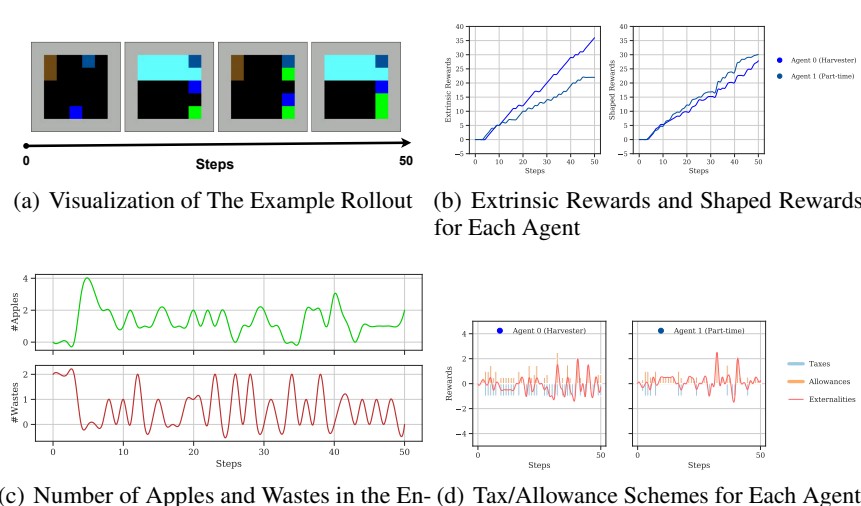

(a) Visualization of The Example Rollout  (b) Extrinsic Rewards and Shaped Rewards for Each Agent

(c) Number of Apples and Wastes in the En- (d) Tax/Allowance Schemes for Each Agent
vironment

Figure 9: An Example Rollout for Cleanup($N = 2$) Environment with A $7 \times 7$ Map.

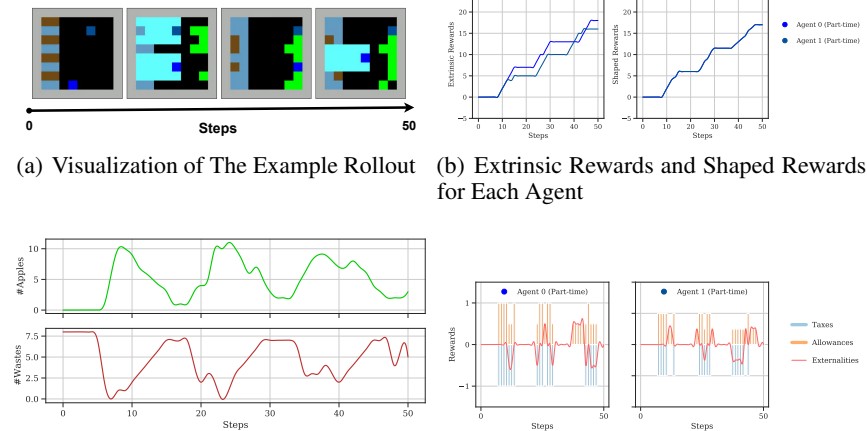

(a) Visualization of The Example Rollout  (b) Extrinsic Rewards and Shaped Rewards for Each Agent

(c) Number of Apples and Wastes in the En- (d) Tax/Allowance Schemes for Each Agent
vironment

Figure 10: An Example Rollout for Cleanup($N = 2$) Environment with A $10 \times 10$ Map.

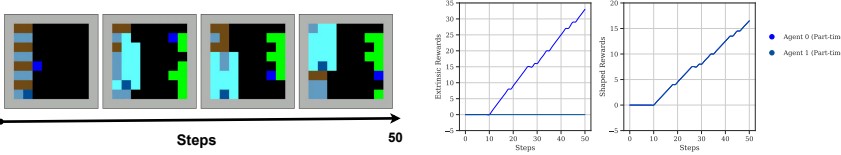

(a) Visualization of The Example Rollout

(b) Extrinsic Rewards and Shaped Rewards for Each Agent

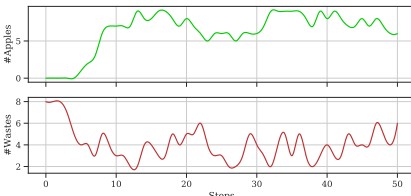

(c) Number of Apples and Wastes in the Environment

Figure 11: Number of Apples and Wastes in the Environment

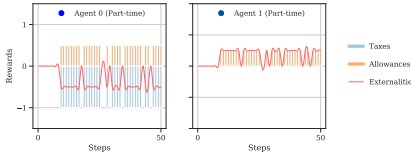

(a) Tax/Allowance Schemes for Each Agent

Figure 12: An Example Rollout for Cleanup($N = 2$) Environment with A $10 \times 10$ Map and Fixed Orientations.

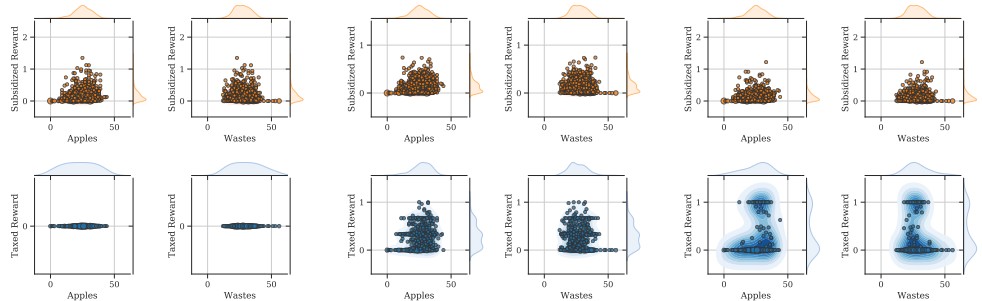

(a) Tax/Allowance Schemes with (b) Tax/Allowance Schemes with (c) Tax/Allowance Schemes with Environmental States for Cleaner Environmental States for Harvester Environmental States for Part-time Agents Agents Agents

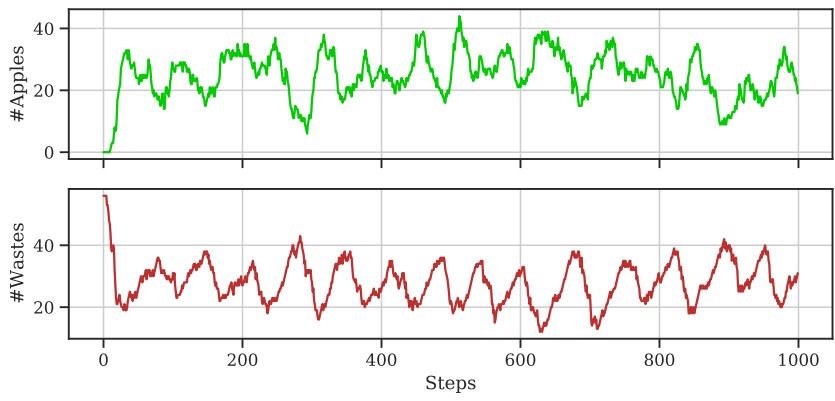

(d) Number of Apples and Wastes in the Environment

Figure 13: An Example Rollout for Cleanup($N = 5$) Environment, supplemental results for Figure 8. (13(a), 13(b), 13(c)) illustrate relationship of environmental states (the number of apples/wastes) and the tax/allowance schemes given by the **LOPT** for 3 types of agents with different socially contributed behaviors. (13(d)) shows the amount for apples and wastes during the episode.

