# OpenReview forum: "LOPT: Learning Optimal Pigovian Tax in Sequential Social Dilemmas"
_NeurIPS.cc/2025/Conference — NeurIPS 2025 poster_

### Official Review · Reviewer_H64g · 2025-06-22

**Clarity:** 3
**Significance:** 3
**Originality:** 3
**Rating:** 4
**Confidence:** 4

**Summary:**

This paper presents Learning Optimal Pigovian Tax (LOPT), a framework to quantitatively address social dilemmas in mixed-motive multi-agent reinforcement learning (MARL). LOPT is inspired from economics, where externalities are effectively modeled into social dilemmas. Specifically, LOPT uses an agent the authors term a tax planner, which learns to allocate allowances to reflect values of externalities. The LOPT framework is validated through theoretical analyses and on MARL benchmarks.

**Questions:**

Q1: It is not clearly mentioned how the LOPT formulation in Sec. 2-3 is different from other widely examined reward shaping approaches, including techniques such as potential-based reward shaping. Could the authors comment?

Q2: A critical result when using methods like potential based reward shaping is that the identity of optimal policies are preserved- i.e., the optimal policy learned when using reward shaping is the same as the optimal policy in the original environment without reward shaping. Does LOPT provide an analogous guarantee?

Q3: While the experimental evaluations are on intuitive environments, the large body of work examining reward shaping and credit assignment in MARL showcase verification on complex multi-agent domains such as StarCraft. Is there a scope for evaluating LOPT in these domains in order to provide a more thorough comparison?

**Ethical Concerns:**

["NO or VERY MINOR ethics concerns only"]

**Final Justification:**

I thank the authors for their detailed response. The comparison of the proposed LOPT with PBRS was especially insightful and offered a perspective that I had not considered. I also appreciate the presenting of additional experiment results demonstrating effectiveness of LOPT. For these reasons, I will increase my original score.

**Limitations:**

Addressed in the Conclusion very briefly. The authors could expand on this, and use a separate Limitations section/ subsection.

**Paper Formatting Concerns:**

None.

**Quality:**

2

**Strengths And Weaknesses:**

Strengths:

(+) The paper is generally well-written and easy to follow.

(+) The notion of introducing externalities from economics to quantify social dilemmas in MARL is novel.

(+) Theoretical analysis to demonstrate that LOPT can effectively quantity externalities.

(+) Intuitive experiment domains to demonstrate the validity of the proposed approach.

(+) Evaluation on multiple baselines showcase improvement of LOPT over other methods.

Areas for improvement:

(-) It is not clearly mentioned how the LOPT formulation in Sec. 2-3 is different from other widely examined reward shaping approaches, including techniques such as potential-based reward shaping.

(-) A critical result when using methods like potential based reward shaping is that the identity of optimal policies are preserved- i.e., the optimal policy learned when using reward shaping is the same as the optimal policy in the original environment without reward shaping. Does LOPT provide an analogous guarantee?

(-) While the experimental evaluations are on intuitive environments, the large body of work examining reward shaping and credit assignment in MARL showcase verification on complex multi-agent domains such as StarCraft. Is there a scope for evaluating LOPT in these domains in order to provide a more thorough comparison?

(-) The readability of Fig. 5 and Fig. 6 can be significantly improved. In the current version, it is impossible to see the legends without zooming in and further, the shaded regions showing variance are difficult to discern.

---

> ### Author Rebuttal · Authors · 2025-07-30
>
> We sincerely thank the reviewer for their detailed feedback and insightful questions. These comments are invaluable for improving our paper. We address the specific points below.
>
> ---
>
> ### To Q1: On the Distinction Between LOPT and Potential-Based Reward Shaping (PBRS)
>
> We thank the reviewer for this crucial question. We acknowledge that this distinction was not made sufficiently clear in the current manuscript, and we will significantly revise the **Introduction** and **Related Work** sections to clarify this point in the final version.
>
> The **fundamental difference** lies in the **objective**. PBRS and its variants are designed to **accelerate the learning process** while guaranteeing that the **optimal policy of the shaped problem remains identical** to that of the original environment (i.e., **policy invariance**). This property is highly desirable for speeding up convergence in standard RL tasks.
>
> However, this very guarantee makes PBRS **fundamentally unsuitable** for resolving **social dilemmas**. The core of a social dilemma is that the optimal policy for **self-interested agents** (i.e., maximizing individual rewards) leads to a **socially suboptimal outcome**. The problem is not inefficient convergence to the optimal policy — the problem is that the **original optimal policy is itself socially undesirable**. Preserving it, as PBRS does, fails to resolve the dilemma.
>
> In contrast, **LOPT is not bound by policy invariance**. Our goal is to **transform the learning problem** itself to find a new, **socially optimal equilibrium**. LOPT achieves this by adding a **learnable reward function (the tax)** designed to bridge the gap between individual agent returns and global social welfare. We formally define this gap as the **externality**. This leads directly to the theoretical guarantees discussed in our response to Q2.
>
> ---
>
> ### To Q2: On Theoretical Guarantees Analogous to Policy Invariance
>
> This is an excellent follow-up question. As discussed above, LOPT intentionally forgoes the **policy invariance** guarantee of PBRS because our goal is to **change the suboptimal equilibrium**, not preserve it.
>
> Instead, LOPT provides a different, more relevant theoretical guarantee for **social dilemmas**: we guarantee that the **learned reward shaping (the tax)** creates a new environment where **agents acting in their own self-interest will converge to a policy that is optimal for collective social welfare**.
>
> Our theoretical results directly support this claim:
>
> - **Theorem 1** establishes the **existence** of an optimal tax function. It proves that there is a tax that can perfectly **internalize the externality**, making the sum of individual rewards in the modified environment **equal to the true social welfare**.
> - **Theorem 2** establishes the **learnability** of this tax, demonstrating that it can be **practically found through our proposed framework**.
>
> In summary:
> While PBRS guarantees **convergence to the original optimal policy**, LOPT provides a guarantee that our method can find a **reward transformation** that makes the **socially optimal policy the new equilibrium** for self-interested agents.
>
> ---
>
> ### To Q3: On Evaluation in More Complex Domains Like StarCraft
>
> We appreciate the reviewer’s suggestion to test LOPT’s scalability and performance in more complex environments.
>
> Our primary motivation for this work was to address **social dilemmas**, which are characterized by a **conflict between individual and collective interests**. Environments like the **StarCraft Multi-Agent Challenge (SMAC)** are primarily **fully cooperative**, meaning agents' interests are aligned. Therefore, SMAC is **not the most suitable benchmark** for evaluating a method designed to resolve social dilemmas.
>
> However, to address the reviewer’s valid concern about **complexity**, we have conducted **new experiments** on the **Harvest** environment, also a widely used environment for social dilemmas. We compare LOPT against strong baselines, **MOA** and **AC**, with **N = 5** agents. After **1.5M training steps**, our results show:
>
> - **LOPT** achieves a stable **social welfare** of approximately **700**.
> - **MOA** achieves a lower social welfare of around **580**.
> - **AC** is highly unstable and only reaches about **550**.
>
> These results demonstrate that **LOPT scales effectively** and significantly **outperforms relevant baselines** in a more complex social dilemma setting.
> Due to the **rebuttal format limitations**, we cannot include the full learning curves here, but we will add a **new appendix section** with these complete experimental results and analysis in the final version of the paper.
>
> **Thank you once again for your valuable and constructive feedback.**

---

> > ### Comment · Reviewer_H64g · 2025-08-03
> > **Thank You Authors**
> >
> > I thank the authors for their detailed response. The comparison of the proposed LOPT with PBRS was especially insightful and offered a perspective that I had not considered. I also appreciate the presenting of additional experiment results demonstrating effectiveness of LOPT. For these reasons, I will raise my original score.

---

> > > ### Author Response · Authors · 2025-08-04
> > >
> > > Thank you very much for your thoughtful feedback and for taking the time to read our response. Your recognition is greatly appreciated.

---

### Official Review · Reviewer_HqTs · 2025-07-01

**Clarity:** 2
**Significance:** 3
**Originality:** 2
**Rating:** 4
**Confidence:** 4

**Summary:**

The paper presents a new model for MARL in which the reward of each agent is shaped to consider social welfare. The new term is inspired by Pigovian tax and is learned using an external agent using gradient descent towards the optimal tax. The paper includes two theorems that together form a theoretical basis for the learned tax to be optimal. Empirical evaluation includes two domains and a variety of agents that are self-interested. The results show that the new method, LOPT, enables the agents to learn a balanced policy that maintains social welfare.

**Questions:**

1. In the prisoner's dilemma, shouldn't all values be negative except the 0s in the original formulation, and except the 8s in the corners in the matrix after the tax?
2. What happens if there is some correlation between x_i and x_j? For example, having everyone use the same power grid simultaneously will cause it to collapse, even though the same usage would not incur such a penalty if it were spread out over time.
3. Given the current explanation for the escape room domain, why would anyone pull the lever?

**Ethical Concerns:**

["NO or VERY MINOR ethics concerns only"]

**Final Justification:**

The authors have responded to my concerns and have provided enough evidence to show they can improve the paper as needed.

**Limitations:**

It can be beneficial to have a short discussion about dual purpose in this research: can someone use the same paradigm as LOPT to *decrease* social welfare to intentionally cause unrest?

**Paper Formatting Concerns:**

Plots are very small and hard to read

**Quality:**

2

**Strengths And Weaknesses:**

Strengths:
+ Looking at shaping social welfare into MARL via a Pigovian tax seems novel
+ The background and the explanations for the new method are clear and informative
+ Empirical evaluation seems sound, and results are consistent with the claims

Weaknesses:
- Notation is inconsistent and sometimes unexplained. For example, in line 159 it seems that the last indexing of r should be (-j) rather than j. Similarly, the notation in line 201 with pi^p_{\Phi_p} but then the same term in Eq. 9 is pi_p^{\Phi_p}. Also, I'm unsure that the numbers in Figure 2 depicting the prisoner's dilemma are correct.
- Theorems are considered a part of the main paper, but the proofs are solely in the appendix, so they cannot be considered a fully sound contribution.
- Related work section is missing on some substantial recent work in the field of solving social dilemmas using MARL, including for example:
1. Li, Yang, et al. "Aligning Individual and Collective Objectives in Multi-Agent Cooperation." The Thirty-eighth Annual Conference on Neural Information Processing Systems.
2. Haupt, Andreas, et al. "Formal contracts mitigate social dilemmas in multi-agent reinforcement learning." Autonomous Agents and Multi-Agent Systems 38.2 (2024): 1-38.
- The explanation of the experimental design is missing and hard to follow if you're not familiar with the domain. For example, can an agent pull the level from every place, or just if they opened the door? Given the current explanation, why would anyone pull the lever?

---

> ### Author Rebuttal · Authors · 2025-07-30
>
> We sincerely thank the reviewer for their time and for providing such detailed and constructive feedback on our manuscript. We are grateful for the recognition of our work's novelty, clarity of explanation, and the soundness of our empirical results. The reviewer’s insightful comments have highlighted several key areas for improvement. We have carefully considered all points and will revise our manuscript accordingly. We will also cite the suggested papers in our related work section.
>
> Below, we address each of the reviewer’s concerns in detail.
>
> ---
>
> ### **1. Notation and Clarity of Figure 2 (Prisoner's Dilemma)**
>
> We thank the reviewer for their careful reading and for identifying inconsistencies in notation and clarity issues in the payoff matrix of the Prisoner’s Dilemma example.
>
> * **Notation:** We acknowledge inconsistencies (e.g., `r_j` vs. `r_{-j}`, and `\pi^p_{\Phi_p}` vs. `\pi_p^{\Phi_p}`), and apologize for the oversight. In our revision, we will unify the notation throughout the main text and appendix, and add a **comprehensive notation table in the appendix** for clarity.
>
> * **Payoff Matrix Clarification:** We appreciate the reviewer’s question regarding the derivation of the after-tax payoff matrix. We realize that our original explanation was insufficiently detailed. Below is the derivation that we will include in the Appendix:
>
> #### **Step-by-Step Derivation:**
>
> 1. **Original Payoff Matrix:**
>
> We start with a classic Prisoner’s Dilemma game:
>
> $$
> \begin{bmatrix}
> (-1, -1) & (0, -10) \\\\
> (-10, 0) & (-8, -8)
> \end{bmatrix}
> $$
>
> Here, mutual cooperation yields \$(-1, -1)\$, and mutual defection yields \$(-8, -8)\$. In the asymmetric case, say (Defect, Cooperate), the defector gains 1 unit (from \$-1\$ to \$0\$), while the cooperator suffers a loss of 9 (from \$-1\$ to \$-10\$), resulting in a **net welfare loss of 8** compared to mutual cooperation:
>
> $$
> (0 + (-10)) - (-1 + -1) = -8
> $$
>
> 2. **Pigovian Correction:**
>
> To internalize the externality, we apply a **Pigovian tax** on the defector equal to the social loss they cause (i.e., 8). Symmetrically, we provide an **equivalent subsidy** of 8 to the cooperator, so the social planner remains revenue-neutral.
>
> 3. **After-Tax Matrix (with Tax + Subsidy):**
>
> $$
> \begin{bmatrix}
> (-1, -1) & (-2, -8) \\\\
> (-8, -2) & (-8, -8)
> \end{bmatrix}
> $$
>
> Explanation:
>
> * In the (Defect, Cooperate) case:
>
>   * Defector: \$0 - 8 = -8\$
>   * Cooperator: \$-10 + 8 = -2\$
>
> * In the (Cooperate, Defect) case: symmetric values.
>
> * Mutual cooperation and mutual defection are unchanged, as no externalities arise in those configurations.
>
> This transformation **shifts the Nash equilibrium** from the socially suboptimal (Defect, Defect) to the socially optimal (Cooperate, Cooperate).
>
> We will add this derivation and both the original and after-tax matrices to the Appendix, along with an intuitive diagram.
>
> ---
>
> ### **2. Correlation Between Agent Actions (e.g., Power Grid Scenario)**
>
> We appreciate this insightful question on the ability of LOPT to handle **correlated externalities** in multi-agent settings.
>
> We confirm that **LOPT is designed to address such scenarios**, where the joint outcome depends on **non-linear and interdependent agent actions**.
>
> In the power grid example:
>
> * The collapse is triggered by specific **joint action profiles** that violate stability constraints.
> * LOPT’s planner optimizes **total social welfare** across all agents, not individual objectives.
> * The tax function \$T(s\_i, a\_i)\$ is conditioned on each agent’s state \$s\_i\$, which can incorporate **global context** or relevant information about others.
> * Through policy gradients based on total welfare, the planner **learns to penalize** harmful joint behaviors and **incentivize coordination** to avoid collapse.
>
> We will **add a paragraph to the experimental analysis and limitations sections** to emphasize LOPT’s capacity to handle correlated externalities, using the power grid domain as a concrete example.
>
> ---
>
> ### **3. Motivation to Pull the Lever in the Escape Room Domain**
>
> Thank you for the opportunity to clarify this scenario. The Escape Room is explicitly designed to create a **moral hazard** involving positive externalities:
>
> 1. **Environment Dynamics:**
>
>    * A large reward is given only to agents that successfully exit the room.
>    * One agent has the option to **open the door** and escape (selfish action) or **pull a lever** located elsewhere (cooperative action) that allows the *other agent* to escape.
>
> 2. **Incentive Misalignment:**
>
>    * **Selfish Action:** Allows one agent to leave but locks the other out (negative externality).
>    * **Lever Pulling:** Enables the *other* agent to escape, but yields no immediate reward to the actor (positive externality).
>
> 3. **LOPT’s Intervention:**
>
>    * Self-interested agents do not pull the lever under the default reward structure.
>    * The planner in LOPT learns to:
>
>      * **Tax** agents who act selfishly (door-opening that blocks others),
>      * **Subsidize** lever-pullers (to compensate for their altruistic action).
>
> As a result, the **reshaped rewards make cooperation individually rational**. The agent who pulls the lever receives a reward from the planner, aligned with the social benefit they generate.
>
> We will **revise the Escape Room experiment description** to clearly articulate this dilemma. The revision will include:
>
> * A step-by-step walkthrough of the environment and reward structure.
> * A diagram to visually illustrate the spatial layout, action possibilities, and social consequences.
>
> ---
>
> Once again, we sincerely thank the reviewer for their valuable and thoughtful feedback. We are confident that these clarifications and planned revisions—on notation, theoretical explanation, and experimental clarity—will significantly improve the quality and accessibility of our paper.

---

> ### Author Response · Authors · 2025-08-01
>
> Apologies, there was a typo in the Original Matrix and After-Tax Matrix (with Tax + Subsidy). It should be revised as follows:
>
> **Original Matrix:**
> |           | Coop         | Defeat       |
> |-----------|--------------|--------------|
> | **Coop**  | (-1, -1)     | (-10, 0)     |
> | **Defeat**| (0, -10)     | (-8, -8)     |
>
> **After-Tax Matrix (with Tax + Subsidy)**
> |           | Coop        | Defeat      |
> |-----------|-------------|-------------|
> | **Coop**  | (-1, -1)    | (-2, -8)    |
> | **Defeat**| (-8, -2)    | (-8, -8)    |
>
> This is because an agent who chooses to cooperate generates a positive externality of +8 and therefore should receive a subsidy of 8. Conversely, an agent who defects imposes a negative externality of -8 and should be taxed 8. We had mistakenly reversed this in the original version. We will also verify whether the same error exists in the main paper and will make corrections as necessary.

---

### Official Review · Reviewer_qvx9 · 2025-07-03

**Clarity:** 3
**Significance:** 3
**Originality:** 3
**Rating:** 5
**Confidence:** 2

**Summary:**

The paper proposes a MARL paradigm to escape the social dilemma in multi-agent sequential games. The method is inspired by Pigovian taxes theory and build the structure of a high-level tax planner into the MARL algorithm. The paper first established the exisitng theory and built its connection with the MARL setting. Then, with the proposed approach, it conducts experiments on classic social dilemma games. The results show that the MARL can successfully escape such local minimal and explains the implication of the tax framework to the games.

**Questions:**

1. Is it possible to extend the algorithm to continuous control space?
2. How is the scalability of the proposed method of like N=20 agents? Does it significantly take more compute?

**Ethical Concerns:**

["NO or VERY MINOR ethics concerns only"]

**Quality:**

3

**Strengths And Weaknesses:**

Strength:

1. The idea of building the Pigovian taxes bias into the MARL algorithm is interesting and the motivation is easy to understand.
3. The proof of concept experiment is comprehensive in analysis and is supportive to the pipeline.

a. The empirical results suggest that the proposed frame work is very strong, e.g, the Escape Room, where the policy is able to converge in just a few steps while the existing baselines struggles.

b. The method has been shown to be able to generalize to more complicated setting including 5 agents and 18x25 size, which suggests certain scalability.

Weakness:

1. Apart from the existing proof of concept experiments, it is unknown of the scalability of the MARL algorithm. For example, with significantly more agents or the control space is larger or no longer discrete. I would imagine that doing similar experiments with significantly more agents and grid size would make the work much more impactful.
2. The paper uses very small figure which make the reader hard to read. i would suggest reducing the narrative and use larger figures and more illustrations including the visualization of the policy of the proposed approach.

---

> ### Author Rebuttal · Authors · 2025-07-30
>
> We sincerely thank the reviewer for their positive feedback and insightful comments. We are encouraged that the reviewer found our idea *"interesting,"* the motivation *"easy to understand,"* and the results *"very strong."* We appreciate the opportunity to clarify the questions regarding the extension to continuous control and scalability.
>
> ---
>
> ### 1. On Extending the Algorithm to Continuous Control Spaces
>
> We thank the reviewer for this insightful question.
>
> Yes, our framework **can be extended to continuous control spaces**. Our decision to focus on discrete action spaces in this work was primarily motivated by the **established benchmarks in multi-agent social dilemma research**. To ensure a fair and direct comparison, we adopted the same experimental settings used by seminal works in this area, including *Learning with Opponent-Learning Awareness (LIO)* and others.
>
> These standard environments—such as the **Cleanup** and **Escape Room** games—are predominantly **discrete**, and demonstrating superior performance on them effectively highlights the **core advantages of our tax-based approach** against existing state-of-the-art methods.
>
> ---
>
> ### 2. On Scalability to a Larger Number of Agents
>
> We appreciate the reviewer raising this important point about **scalability**, which is indeed a critical aspect for any MARL algorithm.
>
> The reviewer is correct that **computational complexity increases** as the number of agents (N) grows. As briefly discussed in the paper (Lines **161–163** and the **caption of Figure 3**), the **primary bottleneck** in our current implementation is the **centralized tax planner**. As *N* increases, the planner’s **observation and action spaces grow**, making its optimization task more challenging.
>
> We explicitly acknowledged this as a limitation in the **conclusion** of our paper. However, we do not see this as a fundamental flaw of the **tax-based paradigm** itself, but rather as a **challenge related to the centralized implementation**.
>
> A promising next step—highlighted as part of our future work—is to develop a **decentralized tax mechanism**. For instance:
>
> - The tax planner could be **decomposed** into multiple, smaller planners operating on **subsets of agents**.
> - Alternatively, agents could learn to **negotiate taxes locally**.
>
> This would **significantly enhance scalability** and is a **core focus of our ongoing research**.
>
> ---
>
> We hope these clarifications have addressed the reviewer's questions.
> We will also ensure that the **figures are enlarged** in the final version for better readability.
>
> **Thank you once again for your valuable and constructive feedback.**

---

### Official Review · Reviewer_jTeR · 2025-07-03

**Clarity:** 3
**Significance:** 2
**Originality:** 3
**Rating:** 4
**Confidence:** 1

**Summary:**

A key challenge in multi-agent settings is how to address social dilemmas - the scenario whereby rational agents acting in their own self interest produce suboptimal social outcomes. The authors aim to address this issue by training an AI agent to apply reward shaping to internalise the social costs produced by each agent, thereby applying a 'pigovian tax'. Under their problem formulation, the authors show the existence of an optimal solution and then demonstrate empirically the performance of a policy gradient method with their propose reward shaping formulation.

**Questions:**

- I may have missed this, but how many runs are each of the results averaged over?
- How many samples are required to develop such a tax planning agent? Is this reasonable in real environments?

**Ethical Concerns:**

["NO or VERY MINOR ethics concerns only"]

**Final Justification:**

The authors have answered my questions and addressed the concerns of other reviewers. I will maintain my score.

**Limitations:**

There is limited discussion on the limitations of their work

**Quality:**

2

**Strengths And Weaknesses:**

Strengths:
- The formulation and proofs of existence seem correct as far as I can tell.
- The experiments compare to a good range of baselines.
-

Weaknesses:
- The paper could be more clearly written

---

> ### Author Rebuttal · Authors · 2025-07-30
>
> We sincerely thank the reviewer for their constructive feedback and valuable questions. Our work presents a **reward shaping method** to solve social dilemmas in MARL, inspired by the economic concept of a **Pigovian tax**. The **"tax planner" agent** acts like a government or regulatory body, learning to apply incentives to steer self-interested agents toward a **collective good**. This framework serves as a practical model for real-world problems like **traffic management** or **carbon pricing**, where a **central planner optimizes social welfare**. We will clarify this framing more explicitly in the revision.
>
> ---
>
> ### 1. How many samples are required to develop such a tax planning agent? Is this reasonable in real environments?
>
> Our tax-planning agent is **highly sample-efficient**. As shown in **Figure 5** and  **Figure 6**:
>
> - It typically **converges within 50,000 (5e4) interaction steps** in the **Escape Room** environment.
> - In the **CleanUP environments**, it converges within:
>   - **10 million (1e7)** steps (N = 2, 7 X 7),
>   - **20 million (2e7)** steps (N = 2, 10 X 10),
>   - **150 million (1.5e8)** steps (N = 5).
>
> This level of sample complexity is **reasonable in real-world settings**, as it corresponds to the **training phase** of a reinforcement learning (RL) agent. Once trained, the **learned policy can be deployed directly** without further sampling. That is, the trained tax planner can take in current observations and **immediately output appropriate incentives**.
>
> To improve clarity, we will **enlarge the learning curve plots** in the revised version.
>
> ---
>
> ### 2. I may have missed this, but how many runs are each of the results averaged over?
>
> We apologize for this omission and have added the missing details in the **Experimental Setup** section of the revised manuscript.
>
> - **Learning Curves**: All quantitative results, including the learning curves, are **averaged over 5 independent runs** using different random seeds.
>   The shaded areas in the plots represent the **95% confidence interval**, demonstrating the **stability** of our method.
>
> - **Policy Analysis**: For qualitative analysis of the **emergent strategy**, we selected the **best-performing policy** from the 5 runs.
>   This approach allows for a **clear and interpretable illustration** of the optimal **Pigovian tax mechanism** discovered by our method.
>
> ---

---

### Decision · Program_Chairs · 2025-09-17

**Decision:**

Accept (poster)

**Comment:**

The paper introduces a novel approach to resolving social dilemmas in MARL, grounded in economic theory and backed by theoretical guarantees.  This work will be of high interest to the MARL, game theory, and AI-for-society communities.There were concerns on clarity and the somewhat narrow scope of the experiments. The authors provide strong rebuttal responses, new results, and clear commitments to improve the final version. I recommend acceptance, conditional on revisions for clarity, expanded related work coverage, and inclusion of the new experimental results.